# Effect of Different Mulch Types on Soil Environment, Water and Fertilizer Use Efficiency, and Yield of Cabbage

Xiaoguo Mu [1,†] , Hu Gao [1], Haijun Li [1,†], Fucheng Gao [1], Ying Zhang [1] and Lin Ye [1,2,*]

1   School of Agriculture, Ningxia University, Yinchuan 750021, China
2   Ningxia Facility Horticulture Engineering Technology Research Center, Yinchuan 750021, China
*   Correspondence: yelin.3993@163.com
†   Xiaoguo Mu and Haijun Li made the same contribution to this paper and should be regarded as co-first authors.

**Abstract:** This study aimed to address the crop growth and development issues caused by environmental factors in the area of the Liupan Mountains in Ningxia. In this area, there is a large temperature difference between day and night due to drought and low rainfall from spring to summer. The effects of farmland mulching for cabbage on soil environment, water and fertilizer use efficiency, and on cabbage were studied by comparing white common mulch (WCM), black common mulch (BCM), white and black biodegradable mulch (WBM and BBM), black permeable mulch (BPM), and black-and-white composite mulch (BWCM). The types of mulch suitable for application in the region were selected after a comprehensive comparative analysis. The results suggested that soil temperature and water content decreased in the mulch of the two biodegradable mulches and the permeable mulch compared with the control (WCM). Meanwhile, soil water content significantly increased into the rainy season in the mulch of BPM. The overall index of soil enzyme activity was 11.8% and 5.2% higher in WBCM and BBM than that in WCM. The soil overall fertility index of WCM exceeded the other treatments by 16.3%, 33.0%, 25.6%, 36.6%, and 25.4%. The water use efficiency and fertilizer bias productivity of BBM and BPM mulch treatments were the highest among all treatments. The economic yield and economic efficiency of cabbage in BBM, BPM, and WBCM mulch treatments were among the best. A comprehensive analysis of the indicators by completing principal components and affiliation functions revealed that WBCM, BBM, and BPM ranked in the top three in comprehensive scores. In conclusion, black biodegradable mulch, permeable mulch, and black-and-white composite mulch can be applied to replace the white common mulch, with black biodegradable mulch treatment performing the best.

**Keywords:** mulch type; soil environment; water and fertilizer use efficiency; yield; economic benefits





## 1. Introduction

Ningxia is a province located in northwestern China that experiences a continental semiarid climate. Xiji County is a typical representative area of this climate, located near the Liupan Mountains. The primary precipitation occurs between July and September, while the dry climate and low temperatures are observed from April to July, which can impact agricultural production [1]. To address this issue, the use of mulching in agricultural production and cultivation can be an effective solution. Mulching helps to improve the soil's water storage and moisture retention capacity, reduces soil water evaporation, and acts as a drought-resistant moisture enhancer. As a result, mulching technology can provide a viable method to improve crop yield and water use efficiency in the region [2,3].

In China, mulching is mainly performed using ordinary black-and-white mulch. However, other types of mulch, such as degradable, permeable, and black-and-white combined ground mulch, have also been used [4,5]. Plain black mulch, as compared to plain white mulch, does not lead to excessively high soil temperature, which can

prevent excessive water and fertilizer consumption and damage to seedlings caused by high temperatures. Additionally, it has a weed control effect and is widely used in agricultural production [6–8]. Black-and-white combined ground mulch is a new type of mulch that combines the functions of black and white mulch. It enhances soil temperature and high reflectivity while also preventing weed control to avoid high-temperature damage to plants and to regulate the root growth environment [9]. In some areas, mulching production results in crop yield reduction, primarily because of an increase in ground temperature and a water deficit [10]. Specifically, the lack of water supply, the low utilization of natural precipitation caused by low rainfall, and the fact that common mulch blocks the soil from receiving rainwater all limit the development of crop production [11,12]. Therefore, relevant researchers have developed permeable mulches [13,14] to overcome this headache. They have the functions of water infiltration, moisture retention, temperature increase, and aeration, and can effectively collect rainwater for infiltration into the mulch [15]. They exert a positive effect on the yield increase of crops [14]. Similar to permeable membranes are degradable mulches, which are biodegradable, non-recyclable, and environmentally friendly membranes [16,17], and which are developed based on the many problems caused by common membranes left in the soil, such as reduced fertility and blocked water and air. Related studies on the application of degradable mulch have suggested that degradable mulch has the same warming and moisture retention effect as ordinary mulch in the early stage of degradation and starts to degrade after a certain period of use, contributing to effectively boosting the soil gas exchange and regulating a suitable soil environment for crop growth [18–20].

To meet the production requirements of high yield and quality, it is challenging to maintain soil water and heat conditions through open field cultivation. In a cabbage production area in Ningxia, five different types of films were evaluated to determine their effectiveness as alternatives to PE films [21]. In comparing the soil physico-chemical properties, enzyme activity, cabbage quality and yield, and economic benefits of the different mulching films, one type demonstrated the best performance in enhancing cabbage economic efficiency and water and fertilizer use efficiency. This mulching film is recommended for broad application in cabbage planting areas in Ningxia to improve agricultural production and soil environment adjustment.

## 2. Materials and Methods

### 2.1. Experimental Site Overview

The experiment was conducted from June to August 2022 in Xiji County (105.73′ E, 35.9′ N), Ningxia. The average elevation in Xiji County is about 2000 m, with significant continental characteristics, mild climate, low rainfall, sufficient light, and a short frost-free period. Additionally, it has 2034.3 h of annual sunshine, 1297.7 mm of annual evaporation, and 350–500 mm of average annual precipitation. During the experiment period, the average rainfall and temperature were 0.9 mm and 18.5 °C. Meanwhile, the precipitation and temperature variability of the test site were large throughout the cabbage reproductive period (Figure 1). The soil pH ranged from 7.58 to 7.81. EC ranged from 582 to 588 mS·cm$^{-1}$. The total N content was 0.6 g·kg$^{-1}$, the alkali-N content was 6.2 mg·kg$^{-1}$, the avail-K content was 23.6 mg·kg$^{-1}$, the olsen-P content was 45 mg·kg$^{-1}$, and the organic matter content of 10.3 g·kg$^{-1}$. The soil fertility was low. The planting method mostly adopted drip irrigation under the mulch.

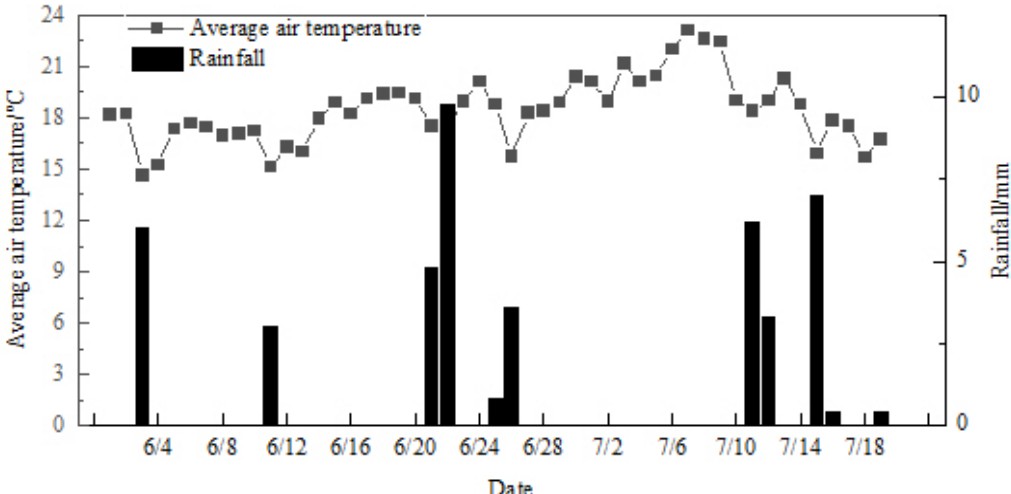

**Figure 1.** Average temperature and rainfall in the test site from June to August. Small weather stations were set up at the trial site to record local rainfall and temperature data.

*2.2. Test Material*

In this experiment, the test cabbage variety was cabbage 15 of China, and healthy seedlings were transplanted for the field trial. Six different types of mulch were purchased directly from domestic mulch manufacturers and the different types of mulch included were WCM, BCM, WBM, BBM, BPM, and WBCM. The precise parameters of the mulches are shown in Table 1. The functional and cosmetic characteristics of the different mulches are shown in Figure 2.

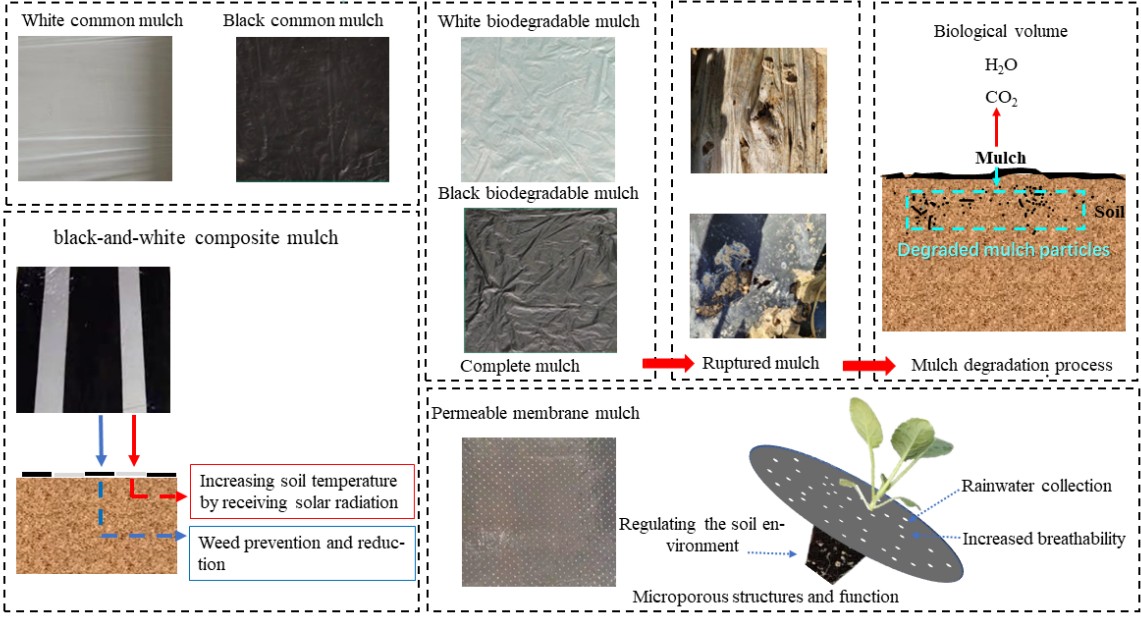

**Figure 2.** The functional and cosmetic characteristics of the different mulches.

**Table 1.** Information parameters of different types of plastic mulch mulches.

| Test Number | Type of Mulch | Main Ingredient | Mulch Thickness/mm |
|---|---|---|---|
| WCM | White common mulch | Polyethylene | 0.010 |
| BCM | Black common mulch | Polyethylene | 0.010 |
| WBM | White biodegradable mulch | Poly(adipic acid)-butylene glycol p-benzenedicarboxylate + poly(lactic acid) | 0.010 |
| BBM | Black biodegradable mulch | Poly(adipic acid)-butylene glycol p-benzenedicarboxylate + poly(lactic acid) | 0.010 |
| BPM | Permeable membrane mulch | Polyethylene | 0.010 |
| WBCM | Black-and-white composite mulch | Polyethylene | 0.010 |

## 2.3. Experimental Design

Six treatments were set up according to the type of mulch, namely white common mulch (WCM), black common mulch (BCM), white biodegradable mulch (WBM), black biodegradable mulch (BBM), permeable mulch (BPM), and black-and-white composite mulch (WBCM). Each treatment was replicated three times with 18 plots of 30 m$^2$ (1 m × 30 m), and each plot was randomly arranged.

Cabbage seedlings of the same age and growth were selected and transplanted on 28 May 2022 with 300 kg·hm$^{-2}$ ternary compound fertilizer as the base fertilizer. The monopoly width was set at 60 cm, the monopoly height was 20 cm, the plant spacing was 30 cm and 40 cm, and the cultivation mode of mulch and drip irrigation was adopted. The total amount of irrigation in all plots is presented in Table 2. The cabbage was harvested on July 24.

**Table 2.** Irrigation time and irrigation amount of cabbage.

| Reproductive Stage | Irrigation Time | Irrigation Amount/m$^3$ |
|---|---|---|
| Seedling stage | 5-30 | 160 |
| | 6-7 | 72 |
| Rosette stage | 6-14 | 68 |
| | 6-22 | 62 |
| Heading stage | 7-4 | 106 |
| | 7-12 | 42 |
| Total | | 510 |

## 2.4. Sampling and Measurement

### 2.4.1. Soil Temperature and Water Content

The soil temperature was measured uninterruptedly in a 24-h cycle. The probe of an RC-4 geothermometer (Jingchuang Electric Co., Ltd., Yongkang, China) was placed at a 15 cm depth in the soil near the root system of the cabbage, the temperature data were recorded every 5 min, and each plot was repeated three times. The base temperature for cabbage growth was 10 °C [22], and temperatures >10 °C were summed up as effective cumulative temperature, while temperatures ≤10 °C were treated as 0. Finally, the effective cumulative soil temperature was calculated for the whole reproductive period. The soil water content was measured at different fertility stages of the cabbage. Eighteen points were randomly selected for each treatment, and the soil around the root system at 0~5 cm and 10~15 cm depth was taken with a ring knife for fresh soil weighing and brought back to the laboratory for drying and weighing.

### 2.4.2. Soil Physicochemical Properties and Enzyme Activity

After harvesting the cabbage, the soil was sampled at a depth of 0-15 cm near the roots of the plants in each treatment, then dried and sieved, and the soil pH was determined using the potentiometer method (water-soil ratio 2.5:1) [23]. PH was measured by the SANXIN MP521 instrument (Xundi Instrument Technology Co., Nantong, China), which has a pH resolution of 0.1/0.01 pH. The nitrogen and phosphorus contents of the soil were determined by Kjeldahl nitrogen determination (Kjeldahl Nitrogen Analyzer Model: KDT-2C) and the molybdenum blue colorimetric method (UV755B scanning type UV spectrophotometer), respectively [23]. The potassium and organic matter contents of the soil were determined by the flame photometric method and potassium dichromate external heating method [23]. The phenol–sodium hypochlorite colorimetric method, 3,5-dinitrosalicylic acid colorimetric method, and potassium permanganate titration method were used for the determination of soil urease, sucrase, and catalase activities [24], and phosphatase activity was determined by the colorimetric method using sodium benzyl phosphate (UV755B scanning type UV spectrophotometer) [23].

### 2.4.3. Water and Fertilizer Use Efficiency

Water and fertilizer use efficiency is expressed in terms of water use rate and fertilizer bias productivity [25] and is calculated as follows:

$$WUE = Y/W \tag{1}$$

$$PFP = Y/F \tag{2}$$

where *WUE* indicates water utilization rate, kg/m$^3$; *Y* is total yield, kg/hm$^2$; *W* is total water consumption, m$^3$/hm$^2$; FPF indicates fertilizer bias productivity, kg/kg; *F* is total fertilizer application, kg/hm$^2$.

### 2.4.4. Yield and Economic Coefficient and Economic Efficiency

Eighteen cabbage plants were selected for yield determination in each treatment, and economic coefficients and economic benefits were calculated as follows:

$$Economic\ factor = economic\ yield/fresh\ bio\ weight \tag{3}$$

$$Economic\ efficiency = total\ revenue - fertilizer\ cost - mulch\ cost \tag{4}$$

### 2.4.5. Soil Overall Enzyme Activity and Fertility Index and Principal Component Analysis

Soil total enzymatic activity index (TEI) referred to the study method of Kong et al. [26]. Soil total soil fertility index (TFI) was calculated by referring to the total enzyme activity index.

$$TEI = \sum Xi/\overline{X} \tag{5}$$

$$TEI = \sum Yi/\overline{Y} \tag{6}$$

where $\overline{X}$ is the mean value of similar enzyme activities and *Xi* is the value of enzyme activity of class *i* of the measured samples. $\overline{Y}$ is the average value of fertility of similar soils and *Yi* is the fertility value of soil type *i* of the measured samples.

Principal component analysis was conducted on soil physical and chemical properties, enzyme activity, economic yield and efficiency of cabbage, and water and fertilizer use efficiency, according to the integrated principal component function model $D = \sum_{j=1}^{n}[u(Xj) \times wj]$, where *u(Xi)* is the contribution ratio, *wj* is the weight, and the integrated principal component values are obtained and ranked.

### 2.5. Data Processing

The SPSS Statistics 21 was used to conduct the analysis of variance (ANOVA). Tukey's HSD test was performed to make multiple comparisons to examine the significant difference between the means of different treatments. Differences were considered statistically significant when $p \leq 0.05$. Experimental results were plotted using Origin 2021.

## 3. Results

### 3.1. Soil Temperature Moisture

Monitoring of soil temperature at a depth of 15 cm below the mulch for the different treatments commenced after seedling transplanting, and from the data recorded (Figure 3c), it was observed that the temperature changed from high to low during the timeframe from the cabbage seedling stage (May 30) to the ripening stage (July 24). Prior to the rosette stage (June 14), soil temperature had a tendency to increase slowly due to atmospheric temperature and solar radiation. After the rosette stage, as cabbage leaf spread increased, the soil received less direct solar radiation and was irrigated on June 14, which caused a sudden decrease in soil temperature. From June 21 to 25, continuous rainy days resulted in another significant decrease in soil temperature. Due to the increasing leaf spread, the soil was able to absorb less heat, which kept the soil temperature in the lower temperature range, and when the intermittent rainfall started on July 11, the soil temperature dropped to below 20 °C.

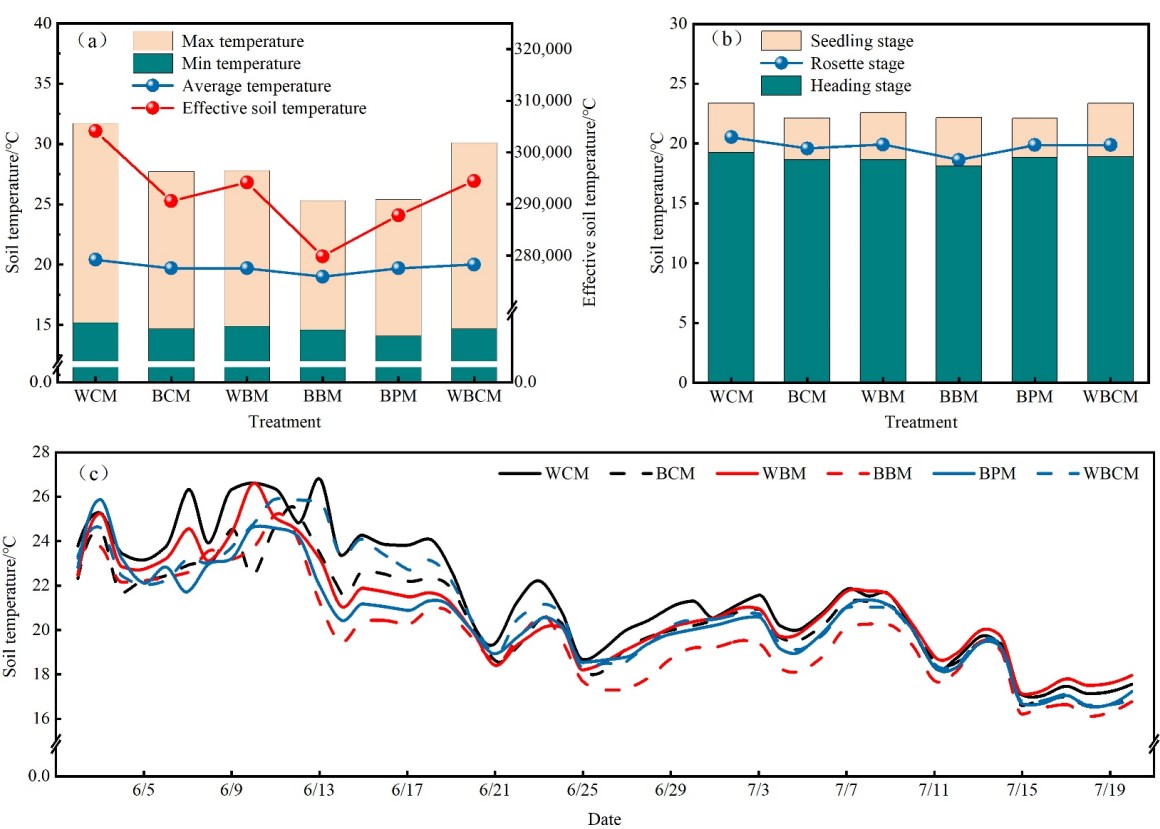

**Figure 3.** (**a**–**c**) Effects of different treatments on soil temperature and effective accumulated temperature.

Soil covered with plain white mulch (WCM) maintained relatively high temperatures throughout the reproductive period, followed by white biodegradable mulch (WBM). The soil temperature covered with black biodegradable and permeable membranes had the lowest temperature compared to the other treatments, which was related to the cracks that appeared later in the degradable membrane and the characteristics of the permeable membrane. Comparing the cabbage fertility stages (Figure 3b), the highest average temperature during cabbage growth was at the seedling stage, and the lowest average temperature was

at the nodulation stage. In terms of the average temperature at the seedling, rosette and nodulation stages, soil temperatures in the WCM, WBM, and WBCM treatments compared with those in the BCM, BBM, and BPM treatments. Similarly, soil temperatures from soil covered with black biodegradable (BBM) and permeable membranes (BPM) were the lowest in all three stages, and the maximum soil temperature was considerably lower than the other four membrane treatments. In addition, after the monitoring of the soil effective cumulative temperature was completed, the effective cumulative temperature of the soil in the WCM treatment was significantly higher than that in the other mulch treatments throughout the reproductive period, and the effective cumulative temperature of the soil in the BBM treatment was significantly lower.

### 3.2. Soil Water Content

As they are dependent on the material and manufacturing process, there are differences in the breathability, water permeability, and water retention of different mulches. From Figure 4, it can be seen that during the four developmental stages of cabbage, the soil water content under the mulch decreases sequentially in time according to the environment, irrigation, and rainfall. During the seedling stage, for transplanting and slowing down the seedlings, the amount of irrigation is higher, and, in order to compare the water permeability of the permeable membrane, the sampling date was the day after the rain (June 4). Therefore, the seedling stage contains relatively high moisture, and the soil water content of the permeable membrane cover (BPM) in this period is higher than the other treatments, especially since the increase in soil water content in the deep layer of 10–15 cm is more obvious.

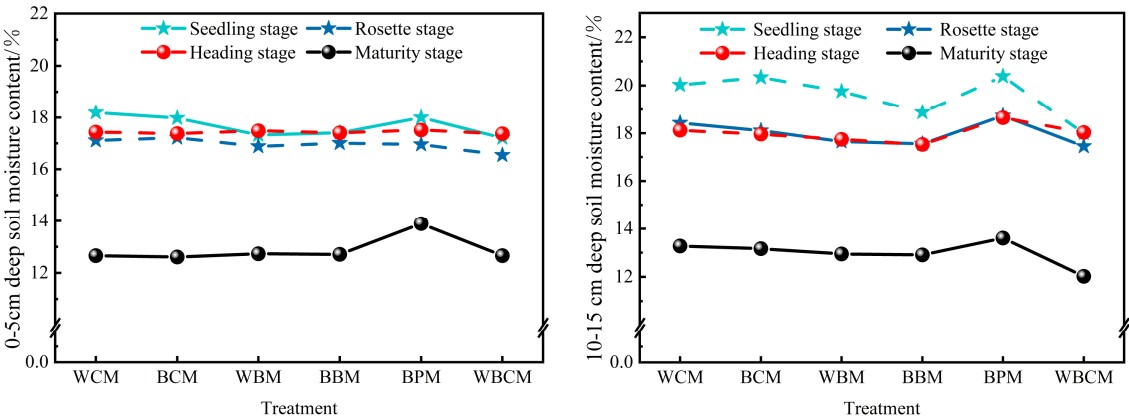

**Figure 4.** Effects of different treatments on soil water content.

The water retention of the degraded mulch is low compared to other treatments. The soil water content in the 0–5 cm deep layer during the rosette and nodulation stages was basically similar, with no significant difference. Large differences in soil water content were found in the 10–15 cm depth layer, but the lowest soil water content was found in the soil covered with two degradable mulches (WBM and BBM), while the soil water content in the soil covered with a permeable mulch (BPM) was significantly higher than the other treatments. Under the condition of stopping irrigation at maturity, the soil water content was greatly reduced, while the soil water content covered by the permeable mulch (BPM) was significantly higher than that of other mulched soil, by about 1.0%. The reason for this is that near the maturity stage, there is continuous rainfall, which is collected by the permeable mulch and infiltrates into the soil.

### 3.3. Soil Chemical Properties

Mulching causes changes in soil physical and chemical properties by altering the macroscopic soil hydrothermal conditions. Figure 5 shows that soil pH was largely maintained between 8.8 and 8.9 and was largely unaffected by the different mulch covers.

However, soil EC values varied due to the effects of different mulches on temperature. Salinity in the soil increased with temperature and so did the EC value. Soil EC was significantly higher in the WCM, WBM, and WBCM treatments than in the other three treatments. The two conventional mulches covered by WCM and BCM had a significant effect on soil organic matter content compared to other types of mulches, with organic matter content of 18.6 g·kg$^{-1}$ and 14.6 g·kg$^{-1}$, respectively, which was considerably higher than the other treatments. Soil total N content was significantly greater in white plain mulch (WCM) than in other mulches, while white biodegradable mulch (WBM) and permeable mulch (BPM) had the lowest soil total N content. The soil fast-acting N content of common white mulch (WCM) was 26.3 mg·kg$^{-1}$, the highest among all mulch treatments, and there was no significant difference in soil fast-acting N content among other mulch treatments. The soil fast-acting phosphorus content of the black and white combined ground mulch (WBCM) was 24.5 mg·kg$^{-1}$, which was similar to that of the two conventional mulches (WCM and BCM) with no significant difference, while the soil fast-acting phosphorus content of the WBM, BBM, and BPM mulch treatments was the lowest, at 19.7 mg·kg$^{-1}$, 20.5 mg·kg$^{-1}$, and 16.5 mg·kg$^{-1}$, respectively. Among all the mulching treatments, the soil with permeable mulch had the highest amount of fast-acting potassium at 68.2 mg·kg$^{-1}$, which was significantly higher than the other treatments, while the soil covered with two conventional mulches (WCM and BCM) had the lowest amount of fast-acting potassium.

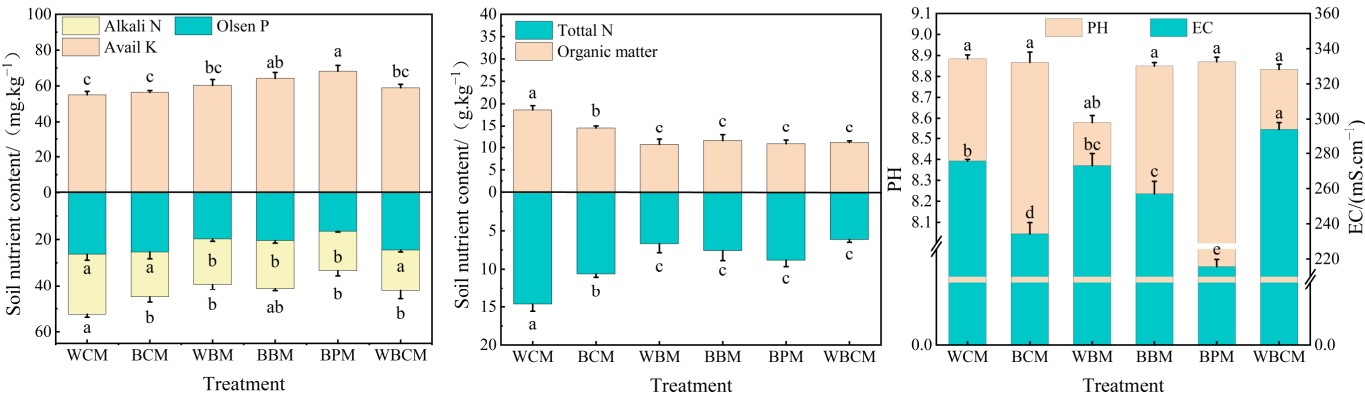

**Figure 5.** Effects of different treatments on soil physicochemical properties. Error bars represent ± SD. Values followed by the same letter within a column in each year are not significantly different at $p \leq 0.05$, as determined by the HSD test.

### 3.4. Soil Enzyme Activity

Soil enzyme activity, as an important indicator of soil fertility, can indirectly reflect soil structure as well as the level of fertility. Figure 6 demonstrates that the activities of various enzymes in the soil changed and formed differences under different mulching treatments. Soil phosphatase and urease activities were significantly increased by 37.4% and 62.2% in the black and white combine ground mulch (WBCM) compared to both WCM and differed from the other overlay treatments to a significant level. Both types of enzyme activity were further increased in the permeable mulch-covered soil (BPM) compared to WCM, but the differences were not significant. The sucrase activity of soil covered with black degradable mulch (BBM) and black and white combination ground mulch (WBCM) was significantly increased by 10.1% and 10.3% compared to WCM, while there was no significant difference in the sucrase activity of soil covered with other types of mulches. The cellulase activity of soil covered with black degradable mulch (BBM) was significantly increased by 15.6% compared to WCM, and the cellulase activity of soil in all other mulching treatments was significantly decreased compared to WCM. Compared to WCM, soil peroxidase activity was significantly higher, by 0.53% and 0.5%, in the black plain mulch (BCM) and in the mulch covered with black and white combination ground mulch (WBCM), while other mulch treatments had similar effects on such enzyme activities to WCM.

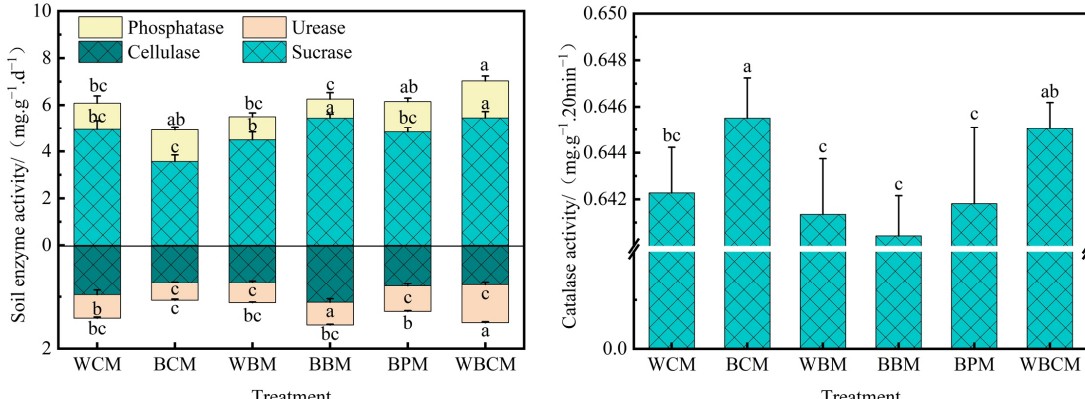

**Figure 6.** Effects of different treatments on soil enzyme activity. Error bars represent ± SD. Values followed by the same letter within a column in each year are not significantly different at *p* ≤ 0.05, as determined by the HSD test.

### 3.5. Water and Fertilizer Use Efficiency, Soil Total Enzymatic Activity, and Fertility Index

The water and fertilizer use efficiencies of different mulching treatments on cabbage are shown in Figure 7. The water use efficiency, as well as fertilizer bias productivity of WCM treatment, was much lower than the other five mulching treatments. In this study, treatments covered with black biodegradable mulch (BBM) and permeable mulch (BPM) had the highest water utilization and fertilizer bias productivity, and they were not significantly different. The highest water and fertilizer use efficiency was found for black biodegradable mulch (BBM) and permeable mulch (BPM), followed by white biodegradable mulch (WBM), black and white combination ground mulch (WBCM) treatment, and black common mulch (BCM), and the lowest water and fertilizer use efficiency was found for common mulch (WCM) from a combination of two indicators: water use efficiency and fertilizer bias productivity. The overall fertility indices of the two common mulch soils were significantly higher than the other mulch types, and the overall enzyme activity indices of the soil covered with black and white mulch and black biodegradable mulch were significantly higher than the other mulch treatments.

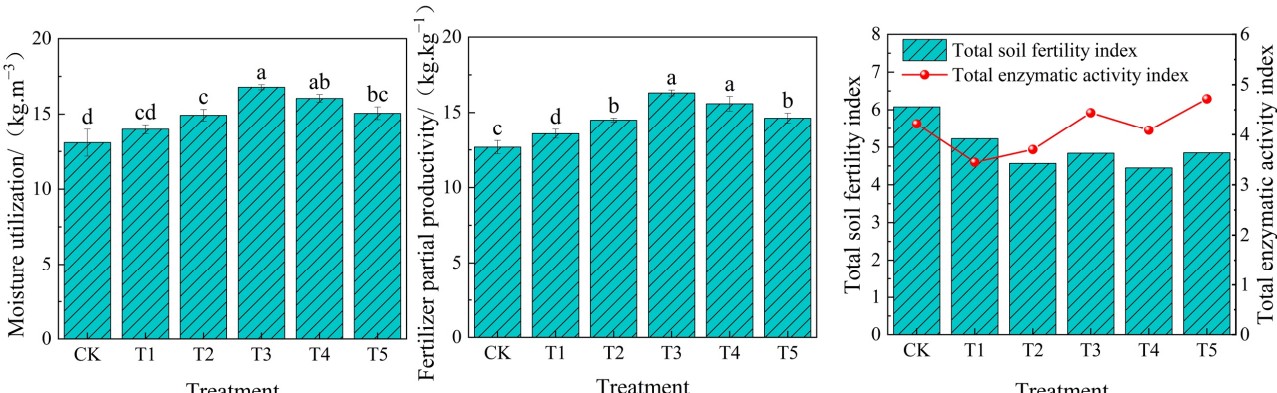

**Figure 7.** Effects of different treatments on water and fertilizer use efficiency Error bars represent ± SD. Values followed by the same letter within a column in each year are not significantly different at *p* ≤ 0.05, as determined by the HSD test.

### 3.6. Yield and Economic Efficiency Analysis of Cabbage

As shown in Table 3, the black biodegradable mulch treatment had the highest economic yield of cabbage, at 89.23 t per hectare, while the remaining five mulch treatments did not show significant differences in economic yield. In terms of economic coefficients, the two coarse mulch (WCM and BCM) treatments had the highest economic coefficients

for cabbage. This reflects that the treatments with higher water and fertilizer use efficiency had lower economic coefficients, as water and nutrients were partly applied to the cabbage leaf bulbs and also partly accelerated leaf growth. Ultimately, the income and cost analysis led to the conclusion that the permeable mulch (BPM) and black biodegradable mulch (BBM) treatments had the highest economic benefits, up to about 27,000 RMB per hectare, and the white biodegradable mulch (WBM) treatment had the lowest economic benefits for cabbage, due to low yields on the one hand and the high cost of the mulch on the other.

**Table 3.** Analysis of different treatments on the yield and economic benefits of cabbage.

| Treatment | Yield (t/ha) | Economic Yield (t/ha) | Economic Coefficient | Total Income/ (Million RMB/ha) | Mulch Cost/ (RMB/ha) | Economic Income/ (Million RMB/ha) |
|---|---|---|---|---|---|---|
| WCM | $99.91 \pm 5.4$ [d] | $75.75 \pm 5.1$ [b] | $0.75 \pm 0.08$ [a] | $3.02 \pm 0.32$ [bc] | 1100 | $2.49 \pm 0.29$ [ab] |
| BCM | $106.97 \pm 4.7$ [c] | $75.81 \pm 7.2$ [b] | $0.71 \pm 0.06$ [a] | $3.04 \pm 0.61$ [bc] | 1820 | $2.43 \pm 0.31$ [ab] |
| WBM | $114.14 \pm 6.3$ [c] | $77.39 \pm 7.5$ [b] | $0.68 \pm 0.11$ [ab] | $3.10 \pm 0.47$ [b] | 4540 | $2.23 \pm 0.42$ [b] |
| BBM | $128.42 \pm 7.4$ [a] | $89.23 \pm 7.0$ [a] | $0.69 \pm 0.10$ [ab] | $3.60 \pm 0.91$ [a] | 4540 | $2.73 \pm 0.18$ [a] |
| BPM | $122.73 \pm 8.7$ [b] | $82.35 \pm 9.1$ [ab] | $0.67 \pm 0.13$ [ab] | $3.30 \pm 0.12$ [ab] | 2000 | $2.69 \pm 0.22$ [a] |
| WBCM | $115.18 \pm 6.6$ [c] | $78.61 \pm 5.2$ [b] | $0.68 \pm 0.09$ [ab] | $3.14 \pm 0.33$ [b] | 2067 | $2.52 \pm 0.36$ [ab] |

The cost of fertilizer and seedlings was RMB 4145/ha for all treatments. Different lowercase letters in the same column indicate significant differences among different treatments ($p \leq 0.05$).

*3.7. Comprehensive Index Score and Evaluation Analysis*

Three principal components were designated by PCA for overall soil fertility, water and fertilizer use efficiency, and economic yield and efficiency of cabbage by taking eigenvalues >1. As can be seen from Table 4, the contribution of PC1 was 59.65%, economic yield, water use efficiency, and fertilizer bias productivity index had a heavy load on PC1 and the contribution of PC2 was 19.11%. Overall soil fertility had the highest load of 0.864 on PC2, and average moisture content and economic efficiency had a larger load on PC3. From this, it can be inferred that cabbage water utilization, fertilizer bias productivity, overall soil fertility, average water content, and economic efficiency mainly influenced the score and evaluation of different treatments.

**Table 4.** Principal component analysis rotation factor loading, eigenvalues, and cumulative variance contribution.

| Project | PC1 | PC2 | PC3 |
|---|---|---|---|
| Total enzymatic activity | $-0.691$ | 0.489 | 0.322 |
| Total soil fertility | 0.437 | 0.864 | $-0.056$ |
| Average temperature | $-0.841$ | 0.349 | 0.059 |
| Average water content | $-0.332$ | $-0.496$ | 0.771 |
| Economic yield | 0.946 | 0.100 | 0.195 |
| Economic benefits | 0.667 | 0.365 | 0.623 |
| Water utilization | 0.987 | $-0.125$ | $-0.022$ |
| Fertilizer partial productivity | 0.986 | $-0.128$ | $-0.025$ |
| Eigenvalues | 4.772 | 1.528 | 1.132 |
| Contribution rate | 59.651 | 19.105 | 14.147 |
| Cumulative contribution rate | 59.651 | 78.756 | 92.903 |

By performing a principal component analysis and calculating the score for each index, the comprehensive evaluation D-values were obtained and ranked by combining the affiliation functions $u(X_1)$, $u(X_2)$, and $u(X_3)$ with weight processing (Table 5). After comprehensive performance analysis in terms of overall soil fertility, enzyme activity, water and fertilizer use efficiency, and economic yield and efficiency, the overlay black and white mulch (WBCM) treatment had the highest overall evaluated D value, followed by

the black biodegradable mulch (BBM) and permeable mulch treatment (BPM). The white biodegradable mulch (WBM), white common mulch (WCM), and black common mulch (BCM) treatments had the lowest overall evaluated D-values. Through the above analysis, the main reason could be the low water and fertilizer use efficiency and economic yield and efficiency.

**Table 5.** Principal components, membership functions, comprehensive evaluation values (D), and rankings for different treatments.

| Treatment | $F_1$ | $F_2$ | $F_3$ | $u(X_1)$ | $u(X_2)$ | $u(X_3)$ | D Value | Sort |
|-----------|-------|-------|-------|----------|----------|----------|---------|------|
| WCM | −1.294 | 0.845 | 0.839 | 0 | 1 | 0.997 | 0.358 | 5 |
| BCM | −0.724 | −0.236 | 0.140 | 0 | 0.564 | 1 | 0.268 | 6 |
| WBM | −0.269 | −0.654 | −1.253 | 0.392 | 0.608 | 0 | 0.377 | 4 |
| BBM | 1.528 | 0.042 | 0.172 | 1 | 0 | 0.088 | 0.655 | 2 |
| BPM | 0.617 | −0.605 | 1.205 | 0.761 | 0 | 1 | 0.641 | 3 |
| WBCM | 0.142 | 0.609 | −1.104 | 0.727 | 1 | 0 | 0.673 | 1 |

## 4. Discussion

Mulching has a significant impact on the soil environment compared to the open field, most notably on soil temperature and moisture. There are differences in soil temperature and moisture changes depending on the type of mulch [27]. In this study, the temperature of the soil covered by permeable mulch was the lowest. With permeable mulch on maize, Shan et al. [28] discovered that the temperature of soil covered by permeable mulch was 1.55 °C lower than that covered by ordinary mulch, owing to the increased ductility of the mulch under higher temperature conditions, micro-opening of stomata, diffusion of high-temperature water vapor outward, and lower temperature under the mulch [29]. Similar to permeable membranes, degradable mulches have similar functions in regulating high temperatures, and holes appear on the surface of the mulch, induced by the environment, and increase in size over time when the soil is covered with degradable mulches, especially in high-temperature environments. These mulches are effective in preventing heat damage to plants caused by high temperatures [30]. Since the soil under the black biodegradable mulch can receive weaker direct solar radiation than the white biodegradable mulch, the soil temperature is lower than the white biodegradable mulch. The effect of black mulch on soil warming was significantly weaker than that of white mulch [31], and the warming effect of the black-and-white combined ground mulch was in between [32], consistent with the results of this study. The soil water content has an effect on root plant uptake and evaporation. In this study, the water content of the soil covered by the two common mulches as well as the black-and-white combined ground mulch was relatively high under low rainfall conditions. Both degraded mulches have a low water content ascribed to the occurrence of cracking situations and the high water vapor transmission through the membrane itself [33]. Among the mulches, soil evaporation is relatively high. This is in line with the case of permeable membranes, where high temperatures take away some of the water to diffuse outward in the form of water vapor under high-temperature conditions through the action of the membrane's structural regulation [29]. Permeable mulch comes into play when entering periods of high rainfall, when it collects rainwater through its micro porous structure to considerably increase the water content of the soil compared to other types of mulch.

Changes in hydrothermal conditions in the soil environment affect nutrient cycling and transport, during which nutrients are transported to be absorbed near the roots after binding with water molecules, leading to differences in soil physicochemical properties due to different amounts of uptake [34]. In this study, the fast-acting phosphorus content of degraded and permeable mulched soils was significantly lower than that of ordinary mulch. Nevertheless, the fast-acting potassium content was significantly higher, consistent with the findings of Wang et al. [35] in tomato, Xie et al. [5] in spring maize, and Liu et al. [36] in cotton. Both degradation and permeable membranes have lower soil fast-acting N contents than overlay and ordinary membranes, in good agreement with the findings of

Xie et al. [5] and other studies. In addition, degradation membranes cover soils with higher fast-acting N content than ordinary membranes [35] due to environmental conditions and differences in the material of the membranes themselves [37]. In the experimental study of degradable mulches, Min et al. [24] revealed an increase in total nitrogen content and no difference in organic matter content in the soil covered by degradable mulches. Nonetheless, Xie et al. [5] suggested that degradable mulches and percolating mulch also contributed to the increase in organic matter content; the crops studied and the test sites were different between the two, resulting in a large difference. In contrast, the present study concluded that both degradable and permeable mulch mulches covered soils had lower total nitrogen and organic matter contents than regular and overlay mulch mulches, and the opposite conclusions reached by the two can be presumed to be due to differences in the soil environment, crop species, and management conditions.

In the present study, soil enzyme activities varied considerably among treatments, indirectly reflecting the altered soil environment due to different mulch coverings. The black biodegradable mulch had a significant promotion of sucrase and cellulase activities in the soil, similar to the results of Liu et al. [38] on maize and Min et al. [24] on wheat. This might be derived from its degradation products and improved soil permeability. The black-and-white combined ground mulch cover soil has elevated urethanogenicity, which is related to the possible species structure given the black-and-white characterization of its membrane. Gu et al. [39] reported that the overlay mulch could raise the temperature as a way to increase the enzyme activity, while too high a temperature could inhibit the enzyme activity. Moreover, their results demonstrated that the special structure of the black-and-white combined ground mulch provided an environment in which the soil urease was more adapted to survive, and therefore the enzyme activity increased. Additionally, the black-and-white combined ground mulch had a significant effect on the elevated phosphatase activity owing to temperature, moisture, and bacterial population [40]. The experimental study of overlay mulch suggested that this may still be associated with the black-and-white structure of the black-and-white combined ground mulch. Therefore, the mechanism of the regulation of the soil environment by the black-and-white combined ground mulch should be further explored. Two common mulches and black-and-white combined ground mulch were covered with higher soil peroxidase activity than degraded and permeable mulches, ascribed to the low temperature of the test site. Sharma et al. [41] revealed that soil peroxidase activity was enhanced during the higher temperature production season. In this experiment, both the permeable membrane and the degraded membrane had holes on the surface that were hollow or cracked, resulting in lower soil temperatures and less enzyme activity at low temperatures compared to the regular mulch membrane with better sealing and insulation properties, as well as the overlay membrane.

Mulch cultivation is effective in improving water and nutrient utilization relative to open fields [42,43]. In a study of corn covered with different mulches, Xie et al. [5] concluded that the water use efficiency and fertilizer bias productivity of biodegradable mulches were higher than those of permeable and ordinary mulches, while the lowest water use efficiency and fertilizer bias productivity were discovered for white common mulch and black common mulch, consistent with the findings of our study. There is little research on water utilization and fertilizer bias productivity of degradable and permeable membranes. This is speculated to be related to the permeability of the membranes. Specifically, permeable and degradable membranes have high permeability and microorganisms in the soil become more active when soil permeability increases [44], which is beneficial for root growth. The difference in water and fertilizer use efficiency also affected the yield of cabbage, contributing to the higher yield of cabbage cultivated with biodegradable and permeable mulch with higher water and fertilizer use efficiency. As suggested in cost and economic benefit calculations, the differences in economic benefits among treatments are not significant. The application of black biodegradable mulch should be promoted from the comprehensive consideration of yield, economic benefits, and soil environmental safety.

## 5. Conclusions

We conducted an experiment in a typical cabbage planting field in Ningxia to test the potential of different types of films to replace PE films. The black-and-white combined mulches and the common ground mulch have comparable soil temperature and water retention rates, while the two biodegradable mulches and the permeable mulch have slightly weaker soil temperature and water retention effects. Permeable membranes utilize the microporous structure of their membranes when entering the rainy season, and then draw rainwater and turn ineffective water into effective water to increase the moisture in the soil. Under the same management conditions, the total enzyme activity of soils covered with permeable mulch, black biodegradable mulch, and black-and-white combination mulch increased, cabbage was able to make full use of irrigation water and nutrients in the soil, and water and fertilizer use efficiency was improved. In particular, the highest economic yield and benefits were achieved with permeable mulch and black biodegradable mulch. Considering the impact of mulch on the soil environment, black biodegradable mulch is undoubtedly environmentally friendly and healthy, with better application prospects. Therefore, this study concluded that black biodegradable mulch can replace conventional mulch to meet local agricultural production needs and promote agricultural development.

**Author Contributions:** Conceptualization, L.Y. and X.M.; methodology, X.M.; formal analysis, F.G.; software, H.L.; validation, H.G. and Y.Z.; investigation, H.L. and Y.Z.; resources, L.Y.; data curation, X.M.; writing—original draft preparation, X.M.; writing—review and editing, X.M.; visualization, H.L.; supervision, H.G.; project administration, L.Y.; funding acquisition, L.Y. All authors have read and agreed to the published version of the manuscript.

**Funding:** This work was supported by [The National Key R&D Program Project] grant number [2021YFD1600302]; [Ningxia Hui Autonomous Region Key R&D Program Project] grant number [2021BBF02019); [Ningxia Hui Autonomous Region Key R&D Program Project Special Project for Attracting Talents] grant number [2021BEB04064]; [Construction of First-class Disciplines in Ningxia Higher Education Institutions] grant number [NXYLXK2017B03]. We are grateful for this support.

**Data Availability Statement:** The authors declare that the data supporting this study are available from the corresponding author upon reasonable request.

**Acknowledgments:** We are very grateful to Lin Ye, for his guidance in writing and editing the paper. We would also like to thank Hu Gao, Hai-jun Li, Fu-cheng Gao, and Ying Zhang for their contribution and assistance in this work. We are particularly grateful to the editors and anonymous reviewers for reviewing the paper and making valuable suggestions to help us improve it.

**Conflicts of Interest:** The authors declare no conflict of interest.

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
