# Peer review of "Effect of Different Mulch Types on Soil Environment, Water and Fertilizer Use Efficiency, and Yield of Cabbage"

_applsci, doi:10.3390/app13074622_

Round 1

Reviewer 1 Report

Changes that are necessary to improve the quality of work.

CONCLUSION NEEDS TO BE IMPROVED, HIGHLIGHTING THE IMPORTANCE OF THE STUDY, SOME DETAILS ARE NEEDED IN THE METHODOLOGY.

Reviewer 2 Report

The manuscript titled "Effect of different mulch types on soil environment, water and fertilizer use efficiency and yield of cabbage" studied the effects of  farmland mulching for cabbage on soil environment, water and fertilizer use efficiency. The manuscript is well written, methods are appropriate and results are appropriately communicated. However, I have provided a few comments that could be incorporated to improve the original version of manuscript. 

Comments to authors:  Abstract: Please avoid long sentences. For example: line 7-11.   Introduction Authors can provide some more details on how this study is unique or novel. I would suggest authors to improve the introduction section particularly the highlighting the novelty of this study (line 66 to 74).    Materials and methods:  Line 84: "The soil was alkaline" please re-write. Line 96: "The appearance characteristics are illustrated in Figure 2." The figure also shows the function of different mulches. Check! Line 166: please check! The font style is confusing.  Line 187: what do you mean by "All data were initially calculated by Excel". Re-write!   Results:   Please use high-resolution figures throughout the manuscript.    Please check the grammatical and structure errors throughout the manuscript. 

Reviewer 3 Report

The article presents the results of research on the Effect of different mulch types on soil environment, water and fertilizer use efficiency and yield of cabbage. The article is relevant, contains the results of reliable experimental studies, has practical significance and scientific originality. To improve the quality of the article, the following changes should be made:

1.) It is necessary to specify the methods for determining humidity and conducting chemical analysis of the soil.

2.) What scientific equipment is used to analyze the chemical properties of the soil after applying various types of mulch?

3.) In the conclusions, the results obtained should be compared with the studies reviewed in the review.

4.) The conclusions should indicate the prospects for the development of further scientific research on this topic.
